# The Concordance of Alveolar Bone Deficiency with Severity of Lip Deformity in Microform Cleft Lip

**DOI:** 10.3390/jcm12010039

**Published:** 2022-12-21

**Authors:** Taehee Jo, Kyehoon Choi, Jaehoon Choi, Junhyung Kim, Kihwan Han, Woonhyeok Jeong

**Affiliations:** Department of Plastic and Reconstructive Surgery, Dongsan Medical Center, Keimyung University College of Medicine, Daegu 42601, Republic of Korea

**Keywords:** cleft lip, congenital healed, bone, anthropometry

## Abstract

Background: We assessed the anthropometric measurements of bone defects in microform cleft lip. Methods: The external phenotypes of the nose and upper lip, and alveolar bone defects in microform cleft lip were measured anthropometrically using multimodal tools and clinical photographs. The height and thickness of the alveolar bone, paranasal hypoplasia, and alveolar volume were measured on CT. Results: Our study included 23 patients with unilateral microform cleft lip. The mean age of the patients was 13.84 ± 12.35 years (range: 1.25–50 years). Alveolar height (C1), thickness (C2), and paranasal hypoplasia (C3) were evaluated on 3D CT scans. The mean differences in C1, C2, and C3 between the cleft and normal sides were 5.52 ± 3.76 mm (*p* < 0.0001), 1.96 ± 2.8 mm (*p* < 0.0001), and 5.57 ± 9.72 mm (*p* < 0.0001), respectively. There was bony deficiency at the cleft side of the alveolar bone and paranasal area. In volumetric analysis, the means of the normal and cleft-side alveolar bone volumes were 6579 ± 2200 mm^3^ and 6528 ± 2255 mm^3^, respectively. The mean difference in alveolar bone volume between the cleft and normal sides was 51.05 ± 521 mm^3^ (*p* < 0.0001). C1 was positively correlated with lip height (F2; correlation coefficient (r) = 0.564, *p* = 0.0051) and dry vermilion thickness (F3; r = −0.543, *p* = 0.0074). The linear regression test revealed significant correlations between C1 and F2 (r^2^ = 0.318, *p* = 0.0051), and F3 (r^2^ = 0.295, *p* = 0.0074). However, there was no correlation between alveolar height and nasal anthropometric measurements. Conclusions: Alveolar bone deficiency was concordant with the severity of soft tissue in microform cleft lip.

## 1. Introduction

Various phenotypes of cleft lip/palate exist [1,2]. Cleft lip is typically categorized as microform, incomplete, and complete based on the extent of the cleft, and as bilateral or unilateral based on its laterality [3]. In an incomplete cleft lip, partial thickness of the lip is preserved, and in a complete cleft lip, a cleft of the whole height of the lip is observed. Microform cleft lip involves partial or total clefting of the orbicularis oris muscle (OOM) without a cleft of the skin and mucosa [4,5]. The clinical features of microform cleft lip are a notched vermilion-cutaneous junction, deficient vermilion on the medial side of the cleft, various extents of cutaneous grooving on the philtral column, and various nasal deformities [6]. The cleft lip affects diverse tissues from skin to muscle, cartilage, bone, and mucosa [7,8]. However, tissue defects are not demonstrated sequentially from skin to mucosa in the spectrum of cleft lip/palate, as cases of incomplete cleft lip with an alveolar cleft or complete cleft palate are not uncommon. Indeed, it has been reported that microform cleft lip is accompanied by a complete cleft alveolus and palate [4]. Similarly, submucous cleft palate is also associated with variable hard palate defects, from the notching of the posterior hard palate to the complete clefting of the hard palate [9].

Most investigations on microform cleft lip have described corresponding surgical techniques or phonotypes [5,6,10,11]. However, there is a lack of studies on whether bone defects actually exist in microform cleft lip. This retrospective study focused on the presence of bone defects and their comparison with anthropometric measurements in microform cleft lip.

## 2. Materials and Methods

This study was approved by the Keimyung University Dongsan Medical Center Institutional Review Board (2022-08-073). The clinical and radiologic records of microform cleft lip patients who were diagnosed at our institution between January 1991 and July 2016 were retrospectively reviewed. Cases with insufficient clinical data, photographs, or radiological data were excluded.

### 2.1. Anthropometric Analysis

We selected materials that were helpful in evaluating the association of soft tissue deformities of the nose and upper lip with alveolar bone defects in microform cleft lip (Table 1). To reflect this clinical tendency as numerical data, using the photographs, the angle of the nostril and the width of the alar base were measured from the worm’s-eye view. The nostril angle (W1) was defined as the angle between the most superomedial point of the nostril and the most inferolateral point of the nostril, which can indicate alar flaring. Then, the nostril angle was calculated using a manual protractor at the cleft and normal sides. The width of the alar base (W2), which was defined as the distance from the vertical line arising from the columella to the alar base of each side, was measured using a ruler. In the frontal view, the height of the philtrum and upper lip, and the thickness of the dry vermilion were measured. The height of the philtrum (F1) was defined as the distance from the columella to the peaks of Cupid’s bow, regarding the notch on the vermilion-cutaneous junction at the cleft side as the counterpart of the peak of the normal side. Thereafter, the vertical distance from the peaks to the horizontal line that crosses the columella was measured as the height of the lip (F2). To measure the thickness of the dry vermilion (F3), the red line, the transition to the oral mucosa, was drawn, and the shortest distance from the red line to both peaks was measured using a ruler. The anthropometric measurement values were calculated as the ratios of the cleft side to the normal side for use in statistical analysis (Figure 1).

### 2.2. Three-Dimensional Computed Tomography Analysis

The height, thickness, and volume of the alveolar bone on each side were analyzed using facial three-dimensional computed tomography, and Mimics and 3-Matics software (Materialise, Leuven, Belgium). The method for measuring alveolar thickness and height was adopted from Meazzini et al. [12]. The alveolar height was the vertical distance from the inferior border of the piriform aperture to the upper boundary of the teeth. It was measured at the point where the distance showed the smallest value at the cleft side and its mirror section on the noncleft side on the frontal view of the three-dimensional reconstructed CT. To measure alveolar thickness (C1), the cross-section that showed the thinnest bony width of alveolar bone among the axial planes was selected. Then, the anteroposterior width of the cortical bone (C2) at the point where the bone was thinnest was measured at the cleft side and its mirror section on the noncleft side. At the lateral aspect, a reduced anteroposterior distance of alveolar bone (C3) may imply paranasal hypoplasia at the affected side. To measure this deformity, we set the most concave point of the alveolar process of the maxilla and coronal facial plane at the anterior aspect of the external auditory canal (EAC) on the lateral view of the three-dimensional reconstructed CT. Then, the shortest distance between the point and the plane was measured bilaterally (Figure 1).

Using Mimics and 3-Matics software, we selected the inner surface area by outlining the cortex of the alveolar bone on the axial CT images and then converted it into a three-dimensional structure. The uppermost axial cross-section was set on the plane of the lowest point of the piriform aperture on the normal side. Measuring down through the alveolar process of the maxilla from the uppermost plane, we set the lowest cross-section on the plane where we could not obviously identify the alveolar bone. To subtract the dental volume within the alveolar bone, the area with tooth-specific density was additionally selected and then removed from the previous outcome. Then, the final structure was divided into two bilateral compartments based on the midline of the central incisors anteriorly, the commissure line of the alveolar bone at the midpoint, and the paranasal sinuses posteriorly. Then, the volume was calculated automatically by the software (Figure 2).

### 2.3. Statistical Analysis

The differences in bony height, thickness, and volume between the cleft side and the noncleft side were statistically analyzed using paired t tests with GraphPad Prism 8^TM^ (GraphPad Software Inc., San Diego, CA, USA). Pearson’s correlation and linear regression analyses were conducted to analyze the correlations of anthropometric measurements with alveolar height and thickness. *p*-values of less than or equal to 0.05 were considered statistically significant.

## 3. Results

Seventy microform cleft lip patients who visited our institution from January 1991 to July 2016 were screened via their medical records, clinical photographs, and computed tomography (CT) scans. Twenty-three patients who had adequate clinical and CT scans were finally enrolled in this study. Forty-seven patients were excluded from this study because they had no pre-operative computed tomography or adequate clinical photographs. Twenty-three patients with microform cleft lip were finally enrolled in this study. The mean age at which the patients underwent photography and radiologic imaging was 13.84 ± 12.35 (1.25 to 50). The male-to-female ratio was 14:9, and the left-to-right ratio was 18:5. None of them had cleft palate or submucous cleft palate. Two patients had medical diseases, one had a ventricular septal defect, and the other had tricuspid regurgitation (Table 2).

Alveolar height (C1), thickness (C2), and paranasal hypoplasia (C3) were evaluated on 3D reconstructed CT scans. The means of the normal and cleft side C1 were 33.98 ± 11.11 mm and 28.46 ± 10.12 mm, respectively. The mean difference in C1 between the cleft and normal sides was 5.52 ± 3.76 mm (*p* < 0.0001). The means of the normal and cleft side C2 were 15.97 ± 9.06 mm and 14.01 ± 7.81 mm, respectively. The mean difference in C2 between the cleft and normal sides was 1.96 ± 2.8 mm (*p* < 0.0001). The means of the normal and cleft side C3 were 173.2 ± 42.67 mm and 167.7 ± 40.29 mm, respectively. The mean difference in C3 between the cleft and normal sides was 5.57 ± 9.72 mm (*p* < 0.0001). There was bony deficiency at the cleft-side alveolar bone and paranasal area (Figure 3). In volumetric analysis, the means of the normal and cleft-side alveolar bone volumes were 6579 ± 2200 mm^3^ and 6528 ± 2255 mm^3^, respectively. The mean difference in alveolar bone volume between the cleft and normal sides was 51.05 ± 521 mm^3^ (*p* < 0.0001; Figure 4).

We performed a correlation test for the bony gaps on CT scans and discrepancies in lip and nose anthropometric measurements between the normal and cleft sides (Table 3). C1 was positively correlated with lip height (F2; correlation coefficient (r) = 0.564, *p* = 0.0051) and dry vermilion thickness (F3; r = −0.543, *p* = 0.0074). In the linear regression test, there were significant correlations between C1 and F2 (r^2^ = 0.318, *p* = 0.0051, Figure 5) and between C1 and F3 (r^2^ = 0.295, *p* = 0.0074; Figure 5). However, there was no correlation between alveolar height and nasal anthropometric measurements.

## 4. Discussion

In our study, microform cleft lip patients demonstrated bony deficiency of the alveolar bone with paranasal hypoplasia on the cleft side compared with the noncleft side. The decreased alveolar height was correlated with a shortening of the lip and vermilion height. However, the bony defects were not correlated with alar flaring or a widening of the alar base.

Yuzuriha and Mulliken [6] minutely categorized microform cleft lip into minor-form, microform, and mini-microform cleft lip based on the extent of notching of the vermilion-cutaneous junction and the severity of nasal deformity. However, few studies have identified the underlying anatomical deformities. Kim et al. [5] reported a decrease in muscle thickness on the cleft side compared with the normal side in minor-form and microform cleft lip patients. Microform cleft lip patients also demonstrated hypoplastic myofibers with nonneurogenic atrophy in a histological study [13]. Although few studies have analyzed OOM defects in microform cleft lip, none have identified the presence of alveolar bone defects in microform cleft lip. Fortunately, we enrolled a patient with a deficiency of alveolar bone and microform cleft lip on the pre-operative CT scan. Thus, we planned to investigate alveolar bone defects in microform cleft lip.

Morphogenesis of the human face starts at the fourth week of embryogenesis. The neural crest is the population of embryonic cells in the margin of the neural folds between the neural and nonneural ectoderm [14]. The facial primordia consist of core mesoderm and the epithelial cover of migrating neural crest cells. Morphogenesis of the upper lip develops from the fusion of the paired maxillary process with the medial and lateral nasal processes originating from the frontonasal prominence. The freely projected margins of the primordia meet each other and are fused by programmed cell death or epithelial–mesenchymal transformation (EMT) from the seventh week of gestation [15]. In this process, the skin, muscle, and skeleton originate from the ectoderm, mesoderm-derived cells, and neural crest-derived facial mesenchyme, respectively [16,17,18]. This difference in origin could provide a clue as to why there was a discrepancy in the degree of tissue defects in microform cleft lip in our results. When the epithelium of the maxillary process and medial nasal process fail to fuse, the development of skin, muscle, and bone clefts is not unexpected because mesenchymal fusion is not possible without the replacement of the epithelial seam with mesenchyme. However, we suppose intact skin could form despite the presence of defects in the underlying tissues of the muscle and skeleton in cases of failure of mesenchymal fusion after intact ectodermal fusion. Until recently, the majority of cleft surgeons recognized that microform cleft lip is a partial or total muscular defect with minor vermilion defects and skin grooves without consideration of bony defects. Although a previous study reported a case of microform cleft lip with a complete alveolar cleft, there have been no original articles investigating the relationship between bone and soft tissue defects until this study [4]. In our study, we identified bony deficiency of the cleft-side alveolar bone and paranasal area with respect to the normal side (Figure 6, Figure 7 and Figure 8). To the best of our knowledge, this is the first report of microform cleft lip with a deficiency of the alveolar bone. Furthermore, alveolar bone height was positively correlated with lip height and negatively correlated with dry vermilion thickness. In other words, the severity of soft tissue defects was concordant with alveolar bone defects. A cleft of the OOM resulted in a shortening of the lip height on the cleft side and thickening of the dry vermilion on the lateral lip [19]. Therefore, considering the outcome, surgeons should evaluate alveolar bone clefts before surgery in short lip-height patients, such as those with the minor form of cleft lip. Additionally, alveolar bone defects always begin superiorly from the pyriform aperture. In view of the processes of embryogenesis, it could be a clue that the OOM could have abnormalities at the superior portion and be associated with nasal deformity. Although we did not find a correlation between the nasal anthropometric measurements and bony deficiency in this study, there may be a meaningful correlation if muscular differences are included. In addition, the clinical data of the retrospective study are limited, and the lack of correlation may be due to the inability of manual measurement tools to detect subtle differences. Thus, future investigations could confirm our hypothesis via an analysis of the anatomy of the OOM around the nasal base using the latest modalities.

A previous study on microform cleft lip focused on surgical techniques and deformity classification [5,6,10,11]. In particular, the OOM flap has been a popular technique for correcting cleft lip nasal deformities in microform clefts. Our study originally reported the relationship between alveolar bone deficiency and nasal deformity. However, the pathogenesis of microform cleft lip is still unclear and has received little attention. Understanding this pathogenesis could inspire new therapeutic approaches. For example, paranasal hypoplasia results in nostril sill depression and posterolateral displacement of the alar base. Therefore, alveolar bone grafting could be a surgical option to correct nasal deformities in microform cleft lip.

Although its retrospective nature is a limitation of our study, this is the first paper to analyze CT scans in microform cleft lip. Our study might serve as a reminder of the importance of bony deficiency affecting lip and nasal deformities in microform cleft lip. In microform cleft lip, the alveolar continuity is always preserved; hence, secondary alveolar bone grafting in the manner in which we perform it in BCLP and UCLP is not needed. Nevertheless, it is reasonable to consider the reconstruction of some bone defects existing in the site of the nasal duct using different surgical methods (e.g., allograft).

## 5. Conclusions

We identified alveolar bone deficiency in microform cleft lip. Alveolar bone height was positively correlated with lip height and negatively correlated with vermilion thickness. Furthermore, alveolar bony deficiency was concordant with the severity of soft tissue defects. Based on our results, we could expect the presence of bony deficiency and develop a new therapeutic modality for this condition in microform cleft lip patients.

## Figures and Tables

**Figure 1 jcm-12-00039-f001:**
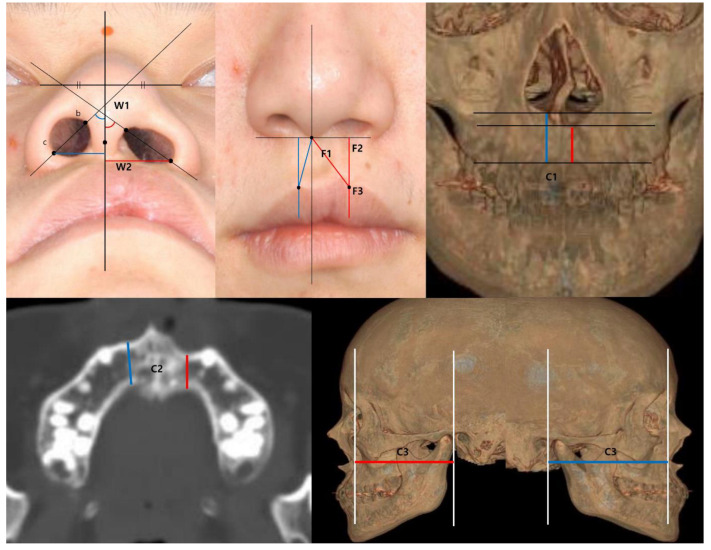
**Anthropometric and three-dimensional analysis.** (**upper, left**) Worm’s-eye view. W1, nostril angle; W2, width of the alar base. (**upper, center**) Frontal view. F1, philtral height; F2, lip height; F3, thickness of dry vermilion. (**upper, right**) Frontal view of the 3D reconstructed CT scan. C1, alveolar height. (**lower, left**) Axial cross-section of the alveolar process. C2, alveolar thickness. (**lower, right**) Lateral view of the 3D reconstructed CT scan. C3, paranasal hypoplasia.

**Figure 2 jcm-12-00039-f002:**
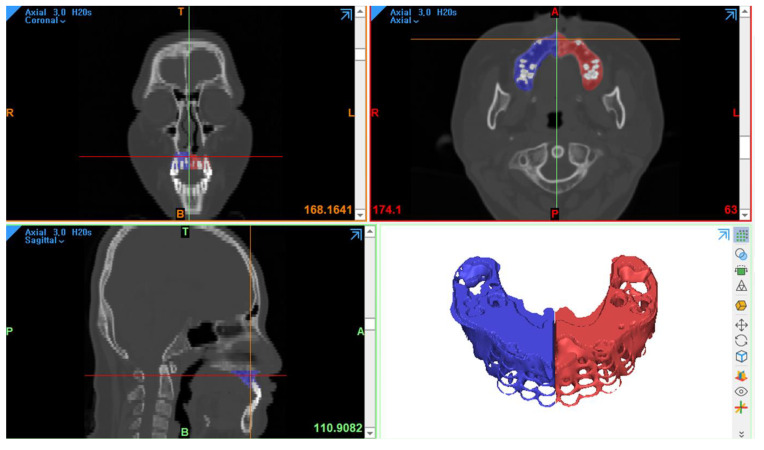
**Volumetric analysis using Mimics and 3-Matics software.** The conjugate of axial images of computed tomography scans is converted to a three-dimensional structure. Then, the volumes of the cleft side (red) and normal side (blue) are calculated automatically by the software.

**Figure 3 jcm-12-00039-f003:**
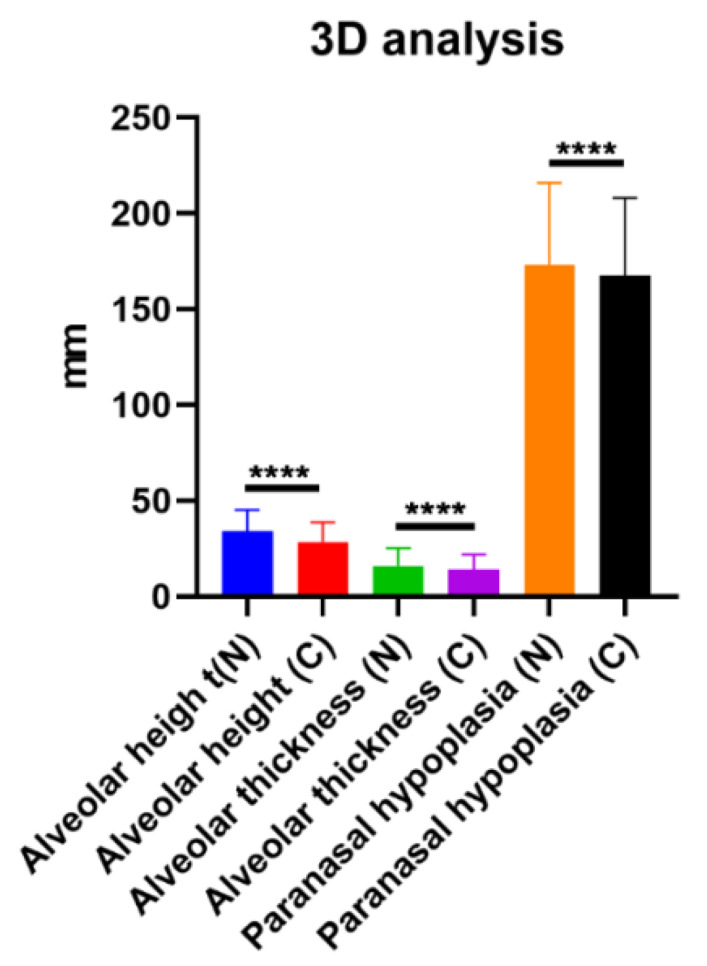
Differences in bony height, thickness, and paranasal hypoplasia between cleft and normal sides. **** *p* < 0.0001.

**Figure 4 jcm-12-00039-f004:**
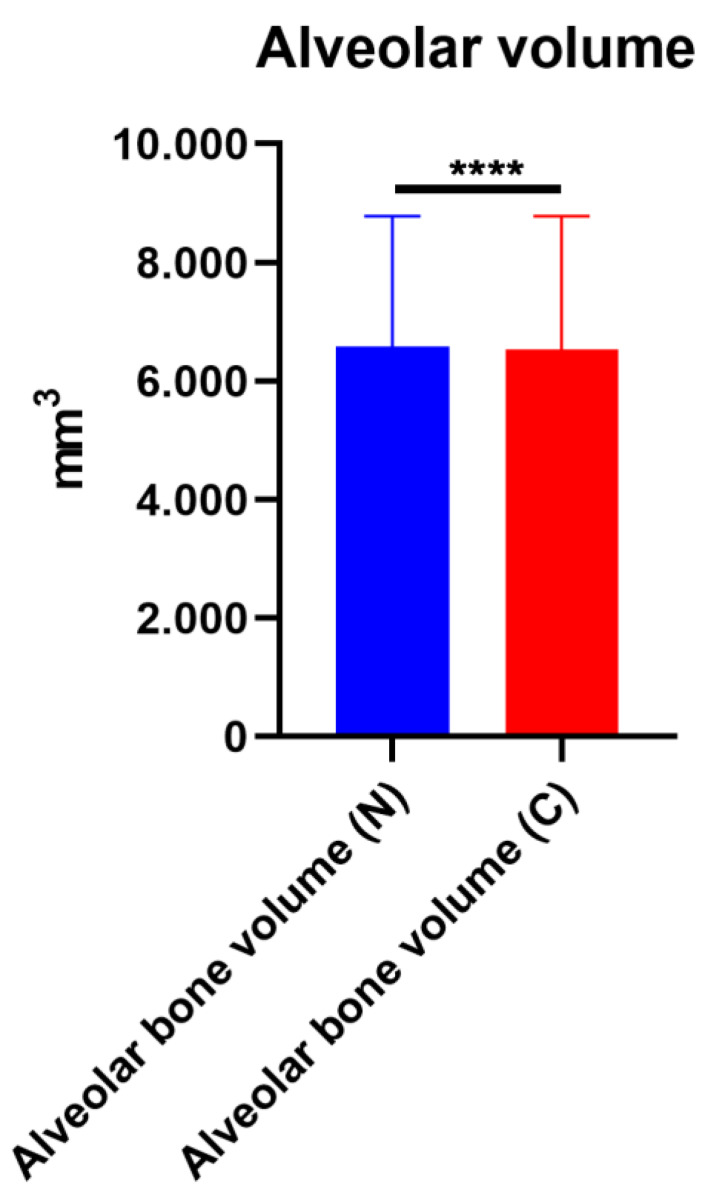
Difference in alveolar bone volume between cleft and normal sides. **** *p* < 0.0001.

**Figure 5 jcm-12-00039-f005:**
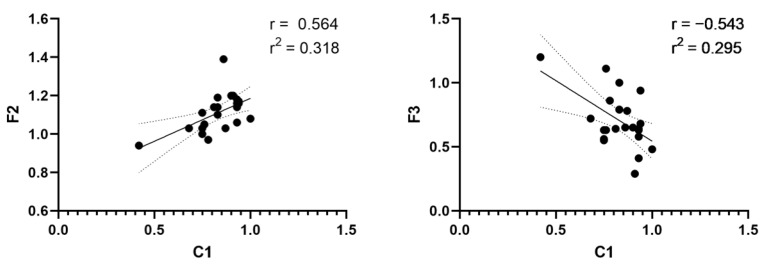
**Correlation and linear regression test.** (**left**) Alveolar height (C1)—lip height (F2) demonstrates a statistically significant positive correlation. (**right**) Alveolar height (C1)—dry vermilion thickness (F3) demonstrates a statistically significant negative correlation.

**Figure 6 jcm-12-00039-f006:**
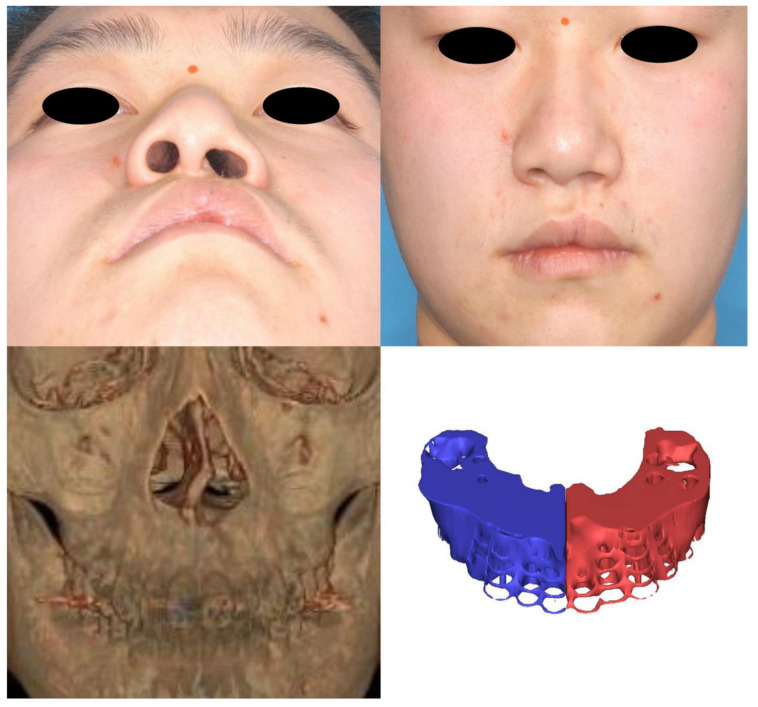
Case 1. A 13-year-old male patient with microform cleft lip. (**upper, left**) Worm’s-eye view. The nostril is flat and wide with a gentle incline on the cleft side compared with the normal side. (**upper, right**) Frontal view. The philtrum is depressed, and a notch is evident on the vermilion-cutaneous junction on the cleft side. The columella deviates to the contralateral side. (**lower, left**) Anterior view of the 3D-reconstructed computed tomography scan. A bony notch on the anterior nasal crest is seen. (**lower, right**) 3D-reconstructed image of the alveolar bone using Mimics software. The alveolar volume was 5453.97 mm^3^ on the cleft side and 5731.04 mm^3^ on the normal side. The volume was smaller on the cleft side, with a difference of 277.07 mm^3^.

**Figure 7 jcm-12-00039-f007:**
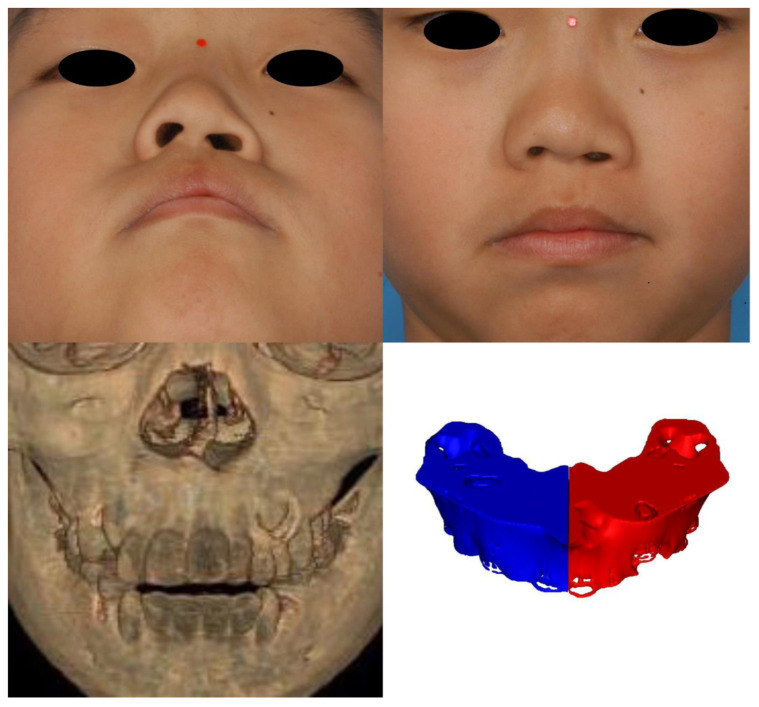
Case 2. A 9-year-old male patient. (**upper**) Worm’s-eye view and frontal view. A typical phenotype of microform cleft lip, an inferolaterally displaced nostril and a notch on the vermilion-cutaneous junction, is shown. (**lower, left**) Anterior view of the 3D-reconstructed computed tomography scan. The anterior nasal spine deviates to the nonaffected side, and the anterior nasal crest is depressed on the affected side. (**lower, right**) The alveolar volume was 6718.81 mm^3^ on the cleft side and 6354.36 mm^3^ on the other side. The volume was larger on the cleft side, with a difference of 364.45 mm^3^.

**Figure 8 jcm-12-00039-f008:**
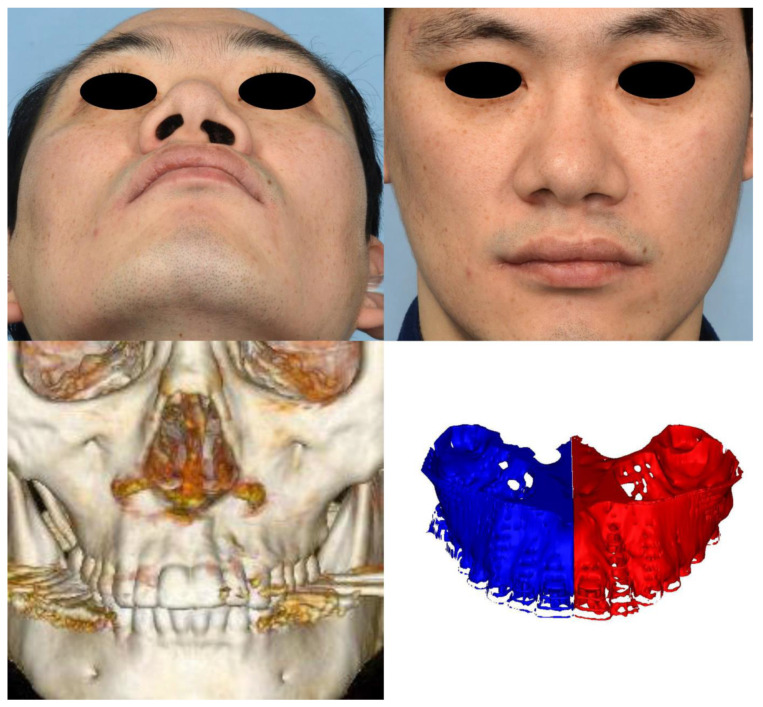
Case 3. A 39-year-old male patient. (**upper**) Worm’s-eye view and frontal view. The clinical phenotypes of the patient were associated with the general characteristics of microform cleft lip. The nostril is more inclined, the distance from the columella to the notch is wide, and the thickness of the dry vermilion is smaller on the cleft side. (**lower, left**) Anterior view of 3D-reconstructed computed tomography scan and 3D-reconstructed image of the alveolar bone using Mimics software. Deviation of the axis is not obvious; however, depression of the anterior nasal crest is well demonstrated. (**lower, right**) The alveolar volume was 9155.08 mm^3^ on the cleft side and 8249.11 mm^3^ on the other side. The volume was larger on the cleft side, with a difference of 905.97 mm^3^.

**Table 1 jcm-12-00039-t001:** Definition of Measurement Items.

Measurement Item	Definition
Nostril angle (W1)	the angle of the vertical line and the line that crosses the nostril apex (the most upper & medial point) and the alar base (the most lateral & lowest point)
Width of alar base (W2)	the shortest distance between the alar base and the vertical line crossing the columella
Philtral height (F1)	the distance between the columella and both peaks of Cupid’s bow (the notch on the cleft side and the peak on the normal side)
Lip height (F2)	the vertical distance from the horizontal line crossing the columella to both peaks
Thickness of dry vermilion (F3)	the distance from the peaks to the redline of the upper lip
Alveolar height (C1)	the distance from the lowest part of the anterior aspect of the piriform aperture to the alveolar bone on the cleft side and the normal side
Alveolar thickness (C2)	the thickness at the thinnest plane of maxillary bone on the axial cross-section of the CT on the cleft side and the normal side
Paranasal hypoplasia (C3)	the distance between the coronal facial plane at the EAC and the most concave point of the maxilla on lateral view of the 3D reconstructed CT

**Table 2 jcm-12-00039-t002:** Patient Demographics.

Characteristic	Value
No.	23
Mean age ± SD, years (range)	13.84 ± 12.35 (1.25 to 50)
Sex	
Male:Female	14:9
Laterality	
Left:Right	18:5
Concomitant disease	
Cleft palate	0
Submucous cleft palate	0
Medical disease	VSD (1), TR (1)
Family history	0

VSD, ventricular septal defect; TR, tricuspid regurgitation.

**Table 3 jcm-12-00039-t003:** Correlation test between bone and soft tissue.

		W1	W2	F1	F2	F3
C1	Correlation coefficient	0.096	0.303	0.377	0.564 **	−0.543 **
	*p*-value	0.664	0.16	0.076	0.005	0.007
C2	Correlation coefficient	0.050	0.101	0.035	−0.021	0.219
	*p*-value	0.821	0.646	0.872	0.926	0.316
C3	Correlation coefficient	−0.164	−0.144	0.078	0.158	0.131
	*p*-value	0.454	0.513	0.722	0.472	0.550

** *p* < 0.01.

## Data Availability

The data presented in this study are available on request from the corresponding author.

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
