# Peer review of "The Concordance of Alveolar Bone Deficiency with Severity of Lip Deformity in Microform Cleft Lip"

_jcm, 2022, doi:10.3390/jcm12010039_

Round 1
Reviewer 1 Report
Journal: Journal of Clinical Medicine
Manuscript ID: jcm-2075995
Type of manuscript: Research Article
Title: The Concordance of Alveolar Bone Deficiency with Severity of Lip Deformity in Microform Cleft Lip
This manuscript was intended to investigate retrospectively the presence of bone defects in microform cleft lip and their correlation with anthropometric measurements.
I was glad to read this well written article focused on the subject that can be often observed in clinical practice of a cleft surgeon, although the most often the association of the microform cleft lip with deficiency of alveolar bone has been denied in literature. The collected material included 23 patients and based on the medical record of patients treated during 25 years.
The problem of reliability of distance measurements on 2D photographs was overcome by the calculations of anthropometric measurement values as the ratios of the cleft side to the normal side for use in statistical analysis. The authors were unable to find a correlation between the nasal anthropometric measurements and bony deficiency. And this can be speculatively attributed to the poor anthropometric analysis making usage of 2D photographs and consequently points out the need of 3D photographs as a part of medical record in the future. It should be included in the part of Discussion describing shortcomings of this study (not only the retrospective character of the study as it is now).
I congratulate the authors, although it appeared to be statistically insignificant, the alveolar volume evaluation and comparison of cleft side and noncleft side – it shows the scale of how subtle differences this article is about.
The last sentence: “Furthermore, various methods for reconstructing hypoplastic bone will be needed as surgical techniques for microform cleft lip patients.” Could be enriched by concluding that: although in microform cleft lip the alveolar continuity is always preserved, hence the secondary alveolar bone grafting in the form as we perform it in BCLP and UCLP is not required, nevertheless the reconstruction of some bone defect existing in the site of nasal duct could be considered using different surgical methods (e.g. allograft etc) .
The sentence: “This result could be explained by an error that occurs in the fusion of the mesoderm after ectodermal fusion.” Dose not derive from the results of this study hence should be deleted.
Author Response
We are grateful for sincere and considerate comments from both reviewers on our manuscript. As both reviewers mentioned, alveolar bone defects are often observed in the clinical setting; however, there is a lack of original articles. It was a pleasure for us to demonstrate the bony defect with 3D images in microform cleft lip patients. We agree about the limitations mentioned. Based on the thoughtful comments, we revised the manuscript and were able to make it more complete. The detailed responses to each comment are outlined below.
- The authors were unable to find a correlation between the nasal anthropometric measurements and bony deficiency. This can be speculatively attributed to the poor anthropometric analysis making use of 2D photographs and consequently points out the need for 3D photographs as a part of medical records in the future. It should be included in the part of Discussion describing shortcomings of this study (not only the retrospective character of the study as it is now).
Response: We definitely agree with your comment and feel regretful not drawing a meaningful result about the relationship between alveolar bone defects and nasal anthropometric measurements. Although there is a possibility that the correlation does not exist, this may be attributed to the limited clinical data, as the study is retrospective and manual tools for measurement, such as a protractor and a ruler, which cannot detect subtle differences. This limitation may be overcome in future studies as the measuring tools are improving and digitalized. In addition, we newly added a mention about it.
- The last sentence: “Furthermore, various methods for reconstructing hypoplastic bone will be needed as surgical techniques for microform cleft lip patients.” Could be enriched by concluding that: although in microform cleft lip the alveolar continuity is always preserved, hence the secondary alveolar bone grafting in the form as we perform it in BCLP and UCLP is not needed, nevertheless the reconstruction of some bone defect existing in the site of nasal duct could be considered using different surgical methods (e.g., allograft etc.).
Response: We appreciate your detailed comments. Surgical techniques are closely related to the understanding of anatomy. However, we just mentioned briefly that the demonstration of hypoplastic bone may lead to better surgical techniques. Referring to your comment, we added a more detailed explanation of the clinical correlations of the results and made the paragraph more elaborate.
- The sentence: “This result could be explained by an error that occurs in the fusion of the mesoderm after ectodermal fusion.” Dose not derive from the results of this study hence should be deleted.
Response: We agree that the sentence should be deleted, and the sentence was revised.
Reviewer 2 Report
This is a meticulously written manuscript on notched lips and their counterpart in the bony skeleton with nothing to improve or criticize. Every cleft surgeon has known or suspected the facts presented, but they have never been a clinical problem. The 3D representation of the hypoplastic gap tooth is interesting. From my point of view, there are no objections to publication.
Author Response
Thank you for your favorable comment on our study.